# Health Outcomes of Children Living in Out-of-Home Care in Metropolitan Western Australia: A Sequential Mixed-Methods Study—A Protocol Paper

**DOI:** 10.3390/children10030566

**Published:** 2023-03-16

**Authors:** Hope Kachila, Caroline Bulsara, Brad Farrant, Alice Johnson, Carol Michie, Charmaine Pell

**Affiliations:** 1Faculty of Nursing and Midwifery, The University of Notre Dame Western Australia, Fremantle 6160, Australia; caroline.bulsara@nd.edu.au; 2Telethon Kids Institute, University of Western Australia, Nedlands 6009, Australia; brad.farrant@telethonkids.org.au (B.F.);; 3Child Protection Unit, Perth Children Hospital, Government of Western Australia Department of Health, Nedlands 6009, Australia; alice.johnson@health.wa.gov.au

**Keywords:** out-of-home care, health processes, health outcomes, mixed methods, children in care

## Abstract

The research protocol described aims to examine and establish the health outcomes of children and young people living in Out-of-Home Care (OOHC) in Perth, Western Australia (WA) from the perspective of the care recipients and service providers. A Study Advisory Panel (SAP) will be established comprised of Aboriginal Elders (because of the over-representation of Aboriginal children in OOHC), health professionals and other relevant stakeholders to help co-design all phases of the study. Mixed methods will be used in data collection and analysis. In the quantitative phase, it is proposed to collect retrospective data from three WA Department of Communities (DOC) districts. The data proposed to be collected includes: the number of children who received initial and annual health assessments, the health needs identified, and interventions put in place to address these needs. The qualitative phase will consist of interviews with service recipients (young people who have exited OOHC and Carers), health service providers (Community Health Nurses, School Nurses, General Practitioners and Paediatricians) and OOHC Case Workers. The research will provide an overview of the current health needs of children and young people in OOHC in WA and the perspectives of these young people, their Carers and service providers on current processes for accessing healthcare. It is anticipated that the study will provide valuable evidence for quality improvement in health service delivery to better meet the health needs of children and young people in OOHC.

## 1. Introduction

Out-of-Home Care (OOHC) is overnight care for children and young people under 18 years of age who cannot live with their families due to child safety concerns [1]. Placing a child or a young person in OOHC is deemed as the last option to provide safety for children and young people who have experienced abuse or neglect by their primary Carers [1]. In Western Australia (WA), the Children and Community Services Act 2004 (CCS Act) [2] is the key legislation that governs OOHC. The government of Western Australia implements the Children and Community Services Act 2004 through the Department of Communities (DOC). The DOC is mandated to provide and coordinate support to families and persons who are either in crisis or at risk, such as children in a violent environment or are victims of abuse [2].

The CCS Act provides definitions of the key terms in this paper. The Chief Executive Officer (CEO) is the head of the DOC. The CEO then becomes the guardian when OOHC commences [1,2]. The CCS Act states that the CEO of the DOC should undertake comprehensive care planning focusing on identifying and meeting the needs of the OOHC population.

There are various OOHC arrangements, guided by the availability of support systems for the affected family, child, or a young person [2]. When a child or a young person has been assigned to a family or person who is paid by the government for caring services, this arrangement is known as foster care. A foster Carer is a person who has expressed interest in caring to the DOC and their qualifications and experience have been assessed and approved by DOC. When a child or a young person is kept in their family, this arrangement is known as kinship care. A family Carer is an approved family member who hosts the child or young person in OOHC. OOHC can also be provided by government-contracted agencies in a house, providing a family-like setup. The contracted agency Carers provide around-the-clock services [2]. Agency agreements and contracts with the government may vary according to the needs. Young people who can live independently may also be provided with housing arrangements suitable for them under the care of the CEO [2].

In 2011, the Commonwealth Government designed a framework to encourage state and territory collaboration to achieve better outcomes for the OOHC population [3]. The development of Australian National Standards for OOHC in 2011 followed [4] and outlined 13 standards that cover critical life domains. Standard five of the Australian National Standards states that children and young people in OOHC are supported to keep healthy by timely comprehensive health assessments and resolution of identified health needs [4,5].

In WA, the DOC produced an OOHC Outcome Measures Framework [6] which provides objective performance measurement of the OOHC system. This framework comprises six outcomes that align with the National Standards for OOHC, of which health is the second highest priority outcome. The health outcome goal is that children and young people in OOHC will have optimal physical, social, and mental health because their initial health needs are assessed and identified on entry to care and managed appropriately. Both the National Standards for OOHC [4] and the WA Outcome Framework for Children and Young People in OOHC [6] state that comprehensive health assessments must be performed for all children and young people entering OOHC and annually for the duration of their stay in OOHC. In addition, the WA DOC Case Worker Manual section 3.4.11 details the process of health care planning for OOHC [7].

Health is one of nine components of care indicated in the WA DOC Case Worker Manual section 3.4.2, which are “safety, care arrangements, health, education, social and family relationships, recreation and leisure, emotional and behavioural development, identity, and culture, and legal and financial” [7].

Working with the CEO to provide services to meet this standard is the OOHC care team. This is a group of individuals who share responsibility for identifying the child or young person’s needs, planning to meet the needs and evaluating the outcomes [1,2]. The group holds regular meetings to review and, revise the care plans, the action of which is aimed at ensuring the best outcome for a child or young person in OOHC [1,2].

The current process outlined in the WA DOC Case Worker Manual: section 3.4.11, Health care planning (Last Modified: 18 November 2022) details the process of health care planning for OOHC [7].

When a child or young person is taken into OOHC, their care automatically becomes the responsibility of the DOC CEO. The DOC Case Workers implement processes on behalf of the CEO. To facilitate health care planning, the DOC Case Workers obtain a new Medicare card and find health and immunisation records. When a child first comes into the CEO’s care, they must have an initial medical assessment (Form 513) within the first 20 days, followed by a more comprehensive health assessment (Form 510) [7]. The completed form 510 is sent to the Department of Health which is expected to schedule the assessment appointment within 30 working days from referral receipt [7]. As part of ensuring that the health of a child or young person in care is not compromised and the care plan has been implemented, a health and development assessment review must be carried out on an annual basis before the care plan is reviewed [7].

A retrospective study by Lima, published in 2018, evaluated life outcomes in the OOHC population in WA and observed poor outcomes compared to their counterparts in the broader population [8]. This study recommended additional state support to improve life outcomes for children and young people in OOHC. Aboriginal children are over-represented in OOHC, and their placement out-of-home potentially adds harm to an already vulnerable population whose health outcomes are poorer than the general Australian population [9,10]. The available evidence suggests that in order to achieve improved health outcomes for those in OOHC, regional-specific data are required [11,12,13,14]. Therefore, research identifying local challenges in OOHC is needed to develop effective recommendations and practical solutions. This study was designed to complement Lima’s study [8] by quantifying the current uptake of the initial and annual health checks for WA children of all ethnicities in OOHC and identifying their health needs. In addition, the study will seek consumer and service provider perspectives on the health outcomes and healthcare processes for this vulnerable group and their recommendations for quality improvement.

### Research Aims

This study protocol aims to determine the health outcomes of children living in OOHC in metropolitan WA from the perspective of the care recipients and service providers. The study has four interrelated objectives:i.To determine what proportion of children and young people in OOHC are recorded in the DOC database as having had the legislatively required initial and annual health assessments.ii.To identify the health needs (if any) as determined by the health assessments and the management of these health needs (including duration before intervention).iii.To explore the experiences of service consumers (people who have left OOHC and past and present Carers of children in OOHC) and past and present service providers (health professionals and OOHC Case Workers) to gain feedback regarding accessing healthcare and to identify enablers and barriers to the current processes along with suggestions for improvement.iv.To analyse the data and make relevant recommendations to stakeholders and policymakers in order to improve the health outcomes of children and young people in OOHC.

## 2. Materials and Methods

### 2.1. Methods and Study Design

#### 2.1.1. Statistics

Statistics indicate that on 30 June 2020, the number of children and young people in OOHC nationally was 44,900; in WA, it was 5498 [15]. Aboriginal children represent approximately 6% of Australia’s children but comprise about 40% of the national total of children in OOHC [15]. In WA, 7% of children are Aboriginal [15], but on 30 June 2020, 57% (2736 of 5498) of children in OOHC were Aboriginal, indicating that in WA, Aboriginal children are significantly over-represented in the OOHC population.

#### 2.1.2. Methodology

The study will employ a four-phase mixed-methods sequential explanatory design [16] within an overarching community participatory action research framework. The study will use co-design principles of mutual exchange and consultation, listening and empathy, flexibility, commitment to the team, and the ability to compromise and collaborate [17].

Participatory Action Research (PAR) principles will facilitate broader, more holistic, and accurate representation and interpretation of the results and hence provide a better basis for implementing recommendations from the study.

A Study Advisory Panel will direct all aspects of this study, including the study methodology, safe community consultation strategies, and study outcome dissemination. This study has stemmed from numerous conversations with Aboriginal and non-Aboriginal stakeholders, including the Study Advisory Panel members. As stated above, the study has engaged Aboriginal leadership because of the over-representation of Aboriginal children in the OOHC population. The study will apply culturally acceptable research methodologies guided by the Noongar worldview and knowledge frameworks to attain community engagement and ensure traditional authority structures are followed [18,19,20].

### 2.2. Phase One—Quantitative Phase

The study proposes to collect retrospective quantitative data from three WA DOC districts. Data will include: the number of children who received initial and annual health assessments, the health needs identified, and interventions put in place to address these needs. Quantitative data will be extracted from the records indicating the number and nature of health assessments conducted by health service providers. This phase aims to determine how many children and young people who came into care between 30 June 2018 and 30 June 2019 and stayed for at least twelve months are recorded as having had initial and annual health assessments, what health needs (if any) were identified, the management of these health needs, and the duration before intervention.

### 2.3. Sample Size and Data Collection

Study sample calculations were based on World Health Organization recommendation on health studies sample size [21]. Assuming the estimated proportion of health needs addressed within 12 months is 25%, a minimum sample size of 114 participant records is sufficient to evaluate the proper population proportion within 5% (absolute precision), with 95% confidence, acceptable limits for the study’s validity.

These data are proposed to be extracted from a de-identified Initial Medical Assessment form (Form 513) and a Comprehensive Health Assessment form (Form 510) and related documents.

Form 510 is completed at entry into the OOHC and used during annual health assessments. The purpose of the assessment is to comprehensively explore the child’s physical, developmental, and psychosocial needs.

The findings of this phase in terms of the health needs of this population will inform the content of the interviews in the second phase.

### 2.4. Data Analysis

SPSS software Version 22 will be used to analyse the data and obtain descriptive statistics. Categorical variables will be summarised as percentages, while continuous variables will be expressed as means and standard deviations.

### 2.5. Phase Two—Qualitative Phase

#### 2.5.1. Methodology

The qualitative descriptive (QD) research method will frame this research phase. QD is a widely used methodology in health research [22] because it facilitates straightforward perspectives from the participants and produces significant data summarised in conversational language [23,24]. Data analysis is driven by the data codes generated during the study. The outcome is a descriptive exploration of experiences and challenges. Hence QD is a preferred method to explore the subjective experiences of young people, Carers and service providers.

Semi-structured individual interviews or focus group discussions will be conducted by the PhD student and key researcher. Participants will be asked about their healthcare experiences, health needs, health services available for OOHC, and the process of accessing the health services, including the challenges, barriers, facilitators, and their feedback and recommendations. Interviews and focus groups will be conducted face-to-face at a suitable location or online ZOOM session as preferred by potential participants. Interviews are estimated to take 50 to 80 min. All interviews will be recorded on a digital voice recorder with participant consent.

Below are some of the questions that will be included in the semi-structured interviews.

Semi-structured Interview Questions: Service Providers (health providers and OOHC Case Workers)

Can you tell me about yourself, your role, and your years of experience in this role providing services to those in OOHC?

We know that children in care experience a range of health issues, but it would really help me to understand what types of health issues are commonest in this population from your own experience.


*Could you tell me more about the commonest types of health issues in the children (in care) whom you have seen (for health-service providers)/case-managed (for OOHC Case Workers?*

*When you see a child or young person in OOHC in your service, do you have all the background information you need to make an informed assessment (e.g., past medical history, previous treatment given)? Do the Carers who attend with the children know about the child’s past medical history? Do the Carers and/or children know why they have come to see you?*

*How long are children and young people in OOHC generally on your waiting list?*
i.
*Could you please describe the process of referral before and after you see a child or young person in OOHC in your service (for health services providers)*
ii.
*Could you please describe the process of referral before and after you see a child or young person in OOHC in your service (for OOHC Case Workers)*


*What is your view about the available health services for children and young people in OOHC?*
i.
*Do you have any recommendations for improvement in health services for these children?*
ii.
*Can you think of one thing or more you would like to see changed that would benefit those in OOHC in future?*




*Thank you for that valuable information. Is there anything else you’d like to add before we end?*


Semi-Structured Interview Questions: Carers


*Can you tell me about yourself, your role, and your years of experience in this role as a Carer in OOHC?*

*We know that children in care experience a range of health issues, but it would really help me to understand what types of health issues are commonest in this population from your own experience.*

*I would like to ask you a few questions about this—is that OK with you?*

*Are you able to tell me a bit about the commonest types of health issues in the children (in care) whom you have cared for?*

*Thinking back to your experiences as a Carer of children and young people in OOHC, what was your experience accessing health care services?*

*What do you think made it easier or harder to access healthcare services?*

*What is your view about the available health services for children and young people in OOHC? In the past? Present?*


Semi-Structured Interview Questions: People who have transitioned from OOHC


*Can you tell me about yourself, the number of years you were in OOHC, and briefly about your experience in OOHC?*

*We know that children in care experience a range of health issues, but it would really help me to understand what types of health issues are commonest from your own experience. I would like to ask you a few questions about this—is that OK with you?*

*It is common for children and young people like yourself to have health issues at some point whilst they are growing up. Could I ask you a few questions about your health? Would that be OK with you?*

*Can you tell me what sort of health problems you have experienced?*

*In light of quality improvement, do you have any recommendations to improve healthcare for those in OOHC? (Accessing health services? How referrals and follow-ups should be done? Documentation?)*

*Can you think of one thing or more you would like to see changed that would benefit those in OOHC in future?*



*Thank you for that valuable information. Is there anything else you’d like to add before we end?*


#### 2.5.2. Sample Size and Data collection

A purposive sampling technique will be used. It is a non-probability sampling approach whereby the participants with the greatest knowledge and experience are selected to enable the researcher to answer the research question [25].

Maximum variation sampling will also be used as it selects from a broad range of potential participants to ensure that the data represents a comprehensive population perspective. Although the numbers are small, the maximum variation approach ensures that the participant data covers a wide range of topics within the research, thereby comprehensively answering the research questions.

The purposive sampling technique will be used to recruit approximately ten participants from four groups. The focus of this phase is to document service consumer’s experiences (people who transitioned from OOHC and past and present Carers) and past and present OOHC service providers (OOHC Case Workers, Community and School health Nurses, General Practitioners and Paediatricians) of the current health processes for children and young people in OOHC and identify the enablers and barriers to accessing healthcare.

The research student will conduct semi-structured interviews. The interviews will enable study participants to elaborate on their experiences and generate comprehensive information regarding this subject [24]. The interview sessions will last 50 to 80 min. Probes and paraphrasing will be used to encourage discussion and clarification of emerging themes [26]. Maximum variation sampling technique will be used to obtain data across demographic variables (e.g., age, number of years in OOHC).

Interviews for young people who have transitioned from OOHC will focus mainly on capturing their personal experiences of managing their health needs and their perceived health outcomes. Interviews with Carers, OOHC caseworkers, and healthcare providers will explore their perceptions of the current health outcomes of this population, challenges, barriers, facilitators, and suggestions for better health services access and provision.

Data collection will be stopped when there is a repetitiveness of themes.

#### 2.5.3. Recruitment

Service providers will refer potential participants by word of mouth and by distributing flyers and information sheets. Due to sensitivities and trust issues, it is best for service providers to act as a contact for potential participants. All interested potential participants will sign and date the study consent form before engaging in study activities.

#### 2.5.4. Qualitative Data Analysis

Interviews will be transcribed from audio recordings and field notes written during the interviews. QSR NVivo software will be used to manage the data analysis of the interviews. Data coding and analysis will generate themes and sub-themes. Deductive and inductive template thematic data analysis will capture personal experiences and minimise researcher interpretation. The thematic analysis will uncover common themes from the transcriptions.

### 2.6. Phase Three—World Café

Phase three will employ the World Café (WC) consensus method with stakeholders, invited participants, and the study advisory panel members. WC is suitable for this study based on participatory research because it supports focused conversations, builds relationships, and establishes knowledge sharing within large and heterogeneous groups [27,28,29]. WC is built on seven integrated principles: (1) background setting, (2) preparation of a hospitable environment, (3) discussing matters in question, (4) getting all participant’s contributions, (5) inviting discussion of various standpoints, (6) identifying various patterns and insights encouraging questions, (7) harvesting and sharing all views of discussions [29].

The study team will set up four to five round tables with up to six participants in a hospitable environment to encourage conversations. Complimentary refreshments will be available for all participants. The interviews and focus group findings will inform WC questions, guiding discussions and generating recommendations. The study team will aim to have three 20-min cycles of uninterrupted conversation. All tables will have a host and will discuss the same question. All participants except the host will move to the following table at the end of each cycle. The host’s role is to brief the incoming group on the question and to allow participants free discussion of their thoughts. In the end, a common pattern emerging from periods of in-depth discussions will be established. Mutual awareness and understanding will grow, and prospects’ next steps will become clearer [29]. A consensus will build on baseline data for quality improvement and future planning of health management services tailored to children and young people in OOHC.

### 2.7. Phase Four—Data Synthesis and Dissemination

Phase four is consolidating and synthesising data collected in phases one to three to identify common themes, draw conclusions, and compile recommendations. Phase one and two data integration will occur during synthesis through (1) narrative, (2) data transformation, and (3) joint displays [30]. The presentation of study data will be a summary describing the identified patterns and themes that emerged from the data collected from all phases of the study, both in broad and narrow contexts of the experiences. Dissemination will be conducted through presentations to stakeholders, community reports, published papers, and conference presentations.

## 3. Discussion

Current health processes are not adequately addressing the health needs of children and young people in OOHC, which compounds poor health outcomes.

It is anticipated that the study findings will provide new and comprehensive data on the health needs of children and young people in OOHC in WA. The study recommendations will be aimed at revising current health processes and improving health service delivery with the aim of optimising the outcomes for children and young people in OOHC. In addition, the study results will be made available as a resource on WA-specific OOHC population health needs. These data will provide substantial evidence for the government to explore more effective models of care and revise policies for this vulnerable population.

### 3.1. Study Feasibility

Community Engagement and Negotiating Partnership: The study has an Aboriginal Elder/Co-researcher on the Study Advisory Panel and support from Aboriginal Elders. The study will apply culturally acceptable research methodologies guided by the Noongar worldview and has received approval from the Western Australian Aboriginal Health Ethics Committee (WAAHEC).

Study methodology: Mixed-methods sequential explanatory design combines the strengths of both quantitative and qualitative research methods, making it a robust methodology. Participatory Action Research (PAR) promotes stakeholder involvement in the research process, encouraging study ownership.

The Study Advisory Panel provides study governance, ensuring that safe research methodologies are implemented. The members include an Aboriginal Elder/Co-researcher, Aboriginal and Non-Aboriginal Carers, the supervisory team, and other OOHC stakeholders.

### 3.2. Study Limitations

DOC source documents are scanned and stored in the DOC database, this study requires access to these documents and manual data extraction. The workload required to extract these data has limited the sample numbers. (1) People who have transitioned from OOHC may be difficult to recruit. (2) The study will focus on OOHC in the Perth metropolitan area due to logistical considerations and, therefore, may not represent the health needs and challenges of the OOHC population in rural and remote areas of WA.

## 4. Conclusions

The study methodology is robust and recommended for research targeting vulnerable populations. The study will gather comprehensive data which can be translated into evidence-based recommendations to address the complex health needs of children and young people in OOHC. The study will increase our understanding of the health needs of children and young people in OOHC in WA. It is anticipated that the results will provide valuable information to researchers, the government (as a service provider and policymaker), stakeholders, and consumers. The results may also become a resource for policy and practice applicable to other jurisdictions working on improving OOHC health outcomes.

## Data Availability

Study Data will be available at the University of Notre Dame Australia.

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
