# Peer review of "Health Outcomes of Children Living in Out-of-Home Care in Metropolitan Western Australia: A Sequential Mixed-Methods Study—A Protocol Paper"

_children, 2023, doi:10.3390/children10030566_

Round 1

Reviewer 1 Report

Abstract: Why do you use the future when all the work happened in the past?

Abstract: “(young people transitioned from OOHC and OOHC Carers)

I don’t understand this explanation.

Abstract: “A Study Advisory Panel will be established comprised of Aboriginal Elders, family members, and other relevant stakeholders which will help co-design all phases of the study”

This sentence appears after the explanation of the study design. It would be better to appear before because we think the study design had their input.

Keywords: you can add more important keywords like ‘mixed methods’ or ‘children at care’

It is unclear if the study embraces only aboriginal or Australian children (all ethnicities). In the Introduction seems that the study is about aboriginal children.

I don’t understand the data presented at the beginning of METHODS AND STUDY DESIGN – STATISTICS. If the study was made in 2018-2019, why do you present data from the population from 2020? This type of statistics can appear in the introduction or when defining the population and sample; however, they should be previous in relation to the study.

An explanation of the over-representation of aboriginal children should be done in the introduction to clarify the role of the aboriginal elders. Then, in the methods section, you present the research procedures.

I cannot understand why you use the future tense in the discussion without presenting any results. Is this an exercise of futurology? The study happened in 2019, and there are no results. Do you need to publish only the protocol?

Why are there two discussion sections?

I think it does not make sense to try publishing (validating) a protocol implemented 3 or 4 years ago. It would be better to publish the study.

Reviewer 2 Report

This study proposed to conduct a sequential mixed-methods study on health outcomes of children living in out-of-home care in Perth metropolitan areas in Western Australia. This four-phase study design which will engage multiple stakeholders, especially aboriginal researchers, is well done since the majority of the study population is aboriginal, and their culture is unique. At this moment, I have several suggestions and/or concerns about the manuscript. My impression of this manuscript is that many sentences could be more specific and need to cite references. In addition, several unnecessary spaces should be removed.

1.      Abstract: The abstract is a bit difficult for readers to understand from lines 12-15. You may also want to mention your research populations regarding definitions of children and young people. It seems to be missing a space before the “The” in line 15.

2.      Introduction:

a.      Suggest spelling out OOHC and DOC in line 28.

b.      Consider briefly introducing what OOHC is and what children and young people in OOHC refer to. Please cite lines 37-38 and this sentence seems off to me.

c.      You might want to explain what OOHC is and be more specific about what and how to improve health outcomes; what health outcomes specifically will be measured.

d.      In lines 52 and 64, you may need to indicate the authors or paper since “[7]” looks odd.

3.      Materials and Methods: Overall, the first part of this section is confusing and could be reorganized. For example, the sentence in line 95 is incomplete, and there is a section titled “DISCUSSION” while there is another discussion in line 234. In addition, there are many unnecessary blank spaces in this part (e.g., lines 197 and 205) whereas some need to have a space (e.g., lines 207 (4) and 208 (6)). You may also want to add a sentence about how participants can drop out from the research anytime and be specific about what kind of semi-structured questions you would ask, who will conduct the interviews, who will be eligible for the qualitative participants, etc. A supplementary of your questionnaire might be helpful for readers to understand.

4.      Discussion: This is relatively short. You could discuss that your population is from metropolitan areas so (i) how your results may be different regarding rurality (i.e., your participants’ demographics may be different); and (ii) how your participant recruitment methods may affect your results (i.e., by word-of-mouth, etc. vs. online survey).

5.      Conclusion: It seems that lines 265-267 should be moved above but not in the last sentence.

6.       References could be better organized (e.g., line 355) and the formats should match the requirements. 

Reviewer 3 Report

The topic of this article is commendable. However, the text has lacks rigour and is therefore vague and incomplete.

Authors should avoid the future tense in the abstract and the text, because they refer to research that has already been done.

The abstract does not have a clear structure or specific results.

The introduction does not help to clarify either the concept or the health criteria taken into account in this research.

Line 114: This is not the place for discussion.

There is a lack of tables displaying data and results.

References (line 296) are incomplete and should be the same font size as the rest of the text.

Round 2

Reviewer 1 Report

The paper is clearer and improved in its structure and content.

Good work.

Reviewer 2 Report

The manuscript has been reconstructed greatly. Two minor suggestions:

1. Some stated responses and the revised text do not match—for example, the following response is inconsistent with the text in the revised manuscript.
d.  In lines 52 and 64, you may need to indicate the authors or paper since “[7]” looks odd.

Response: Thank you for this observation. Sentences 52 and 64 reconstructed to read. A baseline retrospective study evaluating life outcomes in the OOHC population in WA, observed poor outcomes in the OOHC population compared to their counterparts in the broader population. [7]. AND This study will complement earlier studies, such as [7], 

2. Citations need attention to improve. For example, there are a few spaces in #25 reference.

Reviewer 3 Report

The authors have remarkably improved the text clarity.

On the other hand, the explanation given in line 375 and the following should be in the methodology section.

The authors must show examples of the questions that will be part of the semi-structured interviews. My advice is that all these questions should be shown to assess more precisely if the authors fulfil their purpose "to explore more effective models of care and revise policies for this vulnerable population" that they indicate in lines 309 to 311.
